# Anti-Shrinkage Technique for Feline Skin Samples

**DOI:** 10.3390/vetsci11100507

**Published:** 2024-10-15

**Authors:** Caterina Kiniger, Ligita Zorgevica-Pockevica, Simona Vincenti

**Affiliations:** Department of Clinical Veterinary Medicine, Vetsuisse Faculty, University of Bern, 3012 Bern, Switzerland; caterina.kiniger@unibe.ch (C.K.); ligita.zorgevica@unibe.ch (L.Z.-P.)

**Keywords:** skin shrinkage, surgical margins evaluation, fixation technique

## Abstract

**Simple Summary:**

Complete surgical removal of a subcutaneous tumor and its margin evaluation are crucial for preventing local recurrence and avoiding further treatments. However, surgical margins often differ from those identified during histopathological evaluation due to tissue shrinkage. This shrinkage occurs naturally after specimen collection, leading to unreliable therapeutic assessments and inaccurate prognoses. This study aimed to evaluate skin shrinkage in feline cadaver specimens from three different regions and validate a technique to reduce the measurement gap. A significant shrinkage of about one-quarter from the pre-incisional size was observed for all samples. Stretching and pinning the excised tissues on a cork plate with pinpoint needles restored and maintained the original dimensions, even after 48 h of formalin fixation. This simple, fast, and cost-effective technique represents a practical approach to improve skin sample evaluation.

**Abstract:**

Surgical resection of subcutaneous neoplasms with clear margins is crucial for preventing local recurrence and avoiding adjuvant treatments. However, the evaluation of surgical margins often differs significantly from the histopathological assessment due to tissue shrinkage, which can result in inaccurate therapeutic assessments and unreliable patient prognoses. In this study, ten feline cadavers were utilized. Six 50 mm diameter specimens were collected from three regions (thorax, flank, femur) and measured at three time points: T0 (excision time), T1 (10 min after incision), and T2 (at least 48 h after sample collection and formalin fixation). Samples in the study group were stretched and fixed on a cork plate with pinpoint needles after excision to restore their original dimensions. All specimens exhibited a similar trend. After 48 h of formalin fixation, the control specimens showed significant shrinkage, with a reduction of 25.73% in radius and 26.32% in diameter. In contrast, the study specimens demonstrated minimal changes, with a radius reduction of −0.28% and no change in diameter. The results indicate that all feline skin specimens experienced significant shrinkage of approximately one-quarter from their pre-incisional size. Stretching and pinning the excised tissues allowed for the restoration and maintenance of original dimensions even after formalin fixation. This technique represents a valid and practical approach to minimize tissue shrinkage.

## 1. Introduction

The primary treatment for any subcutaneous neoplasia is, whenever possible, surgical resection, and the follow-up care depends on the operation’s outcome in achieving proper clear margins. Clear margins are obtained when a specific distance (usually in mm or cm) is maintained from the neoplastic cells to the resected margins; therefore, they are identifiable only by a histopathological evaluation [1]. If clear margins are not obtained, the prognosis worsens, necessitating the initiation of additional treatments like chemotherapy or radiotherapy, when feasible. Ensuring clear margins is crucial for improving outcomes and reducing the need for more aggressive interventions in patient management and care [1,2]. For this reason, both the creation and accurate evaluation of the specific annulus surrounding the tumor are crucial for the patient’s prognosis. A well-defined annulus can significantly impact treatment decisions, ultimately influencing the effectiveness of interventions and improving overall patient outcomes.

Tissue shrinkage is a well-documented and known process, which notably alters the size of the tissue once excised and therefore leads to an evident discrepancy between the surgical and the histopathological measurements [3]. While this phenomenon is widely acknowledged in the medical literature, a lack of clear practical recommendations remains to mitigate its impact in the everyday clinical setting, leaving clinicians without definitive strategies to address this challenge effectively. The aim of our study was to evaluate the effectiveness of a new, cost-effective technique, to avoid skin shrinkage. We hypothesized that our technique would permit us to maintain the original sample dimensions even after formalin fixation, resulting in an effective and efficient new sampling method.

## 2. Materials and Methods

### 2.1. Animals

Ten domestic shorthair cat cadavers were included in the study. All the cadavers were donated for research purposes. Following euthanasia, each cadaver was immediately frozen at −20 degrees Celsius to preserve tissue integrity. After a few days, the cadavers were carefully thawed at room temperature, specifically at 18 degrees Celsius, to prepare them for the study.

The data analyzed for the inclusion in this project comprised the signalment and general anamnesis, together with the medical history and comorbidities of the patients, with particular attention to skin diseases. The reason for admission, any therapies, and the reason for euthanasia were considered. The presence of tumors of the skin, lesions, skin diseases, or macroscopically visible anomalies were exclusion criteria. Therefore, only cat cadavers without any known or visible skin alterations were taken into consideration for the study.

The work described in this manuscript involved the use of non-experimental (owned or unowned) animals and procedures that differed from established internationally recognised high standards (“best practice”) of veterinary clinical care for the individual patient.

### 2.2. Procedure

To harvest the samples, the cadavers were placed in lateral recumbency, and bilateral sections from the thorax, flank, and femur regions were selected. A total number of six samples from each cat were obtained (Figure 1). Thorax samples were taken immediately caudal to the forelimbs, flank samples were selected immediately posterior to the ribs, and femur samples were collected from the hind leg at approximately midthigh. All the harvested samples included the subcutis and underlying fascia.

All the samples were circular in shape and their pre-excisional diameter was 50 mm.

Premade bendy plastic 50 mm templates were used to mark the excisional areas with skin markers. Before sampling, a coin was tossed to determine which side the samples should belong to, whether to the study group or to the control group.

After drawing the excision area on the cat skin, the diameter was controlled with a Digital Calliper (reading 0.01 mm). The center of the area was marked in the middle of the drawn section, and a vertical and a horizontal line were traced to delineate and mark the craniocaudal and ventrodorsal cardinal points.

A skin incision was made along the drawn line through the skin, subcutis, and superficial fascia. Immediately after, the dorsal margin of the samples was marked with a single interrupted suture and the cranial margin with two simple interrupted sutures. To complete the surgical excision, dissection of the deep fascial margin was performed and immediately after, the first measurements were performed. First, the radius of the samples was measured from the epidermis side in the four cardinal points in a clockwise direction for the right side of the cadavers, and in an anticlockwise direction for the left side of the cadavers. The first measurement went from the center to the dorsal edge of the specimen and was named “D” (dorsal). The second measurement went from the center of the sample in the caudal direction and was named “Ca” (caudal). The third measurement went from the center to the ventral edge of the sample and was named “V” (ventral). The last measurement went from the center to the cranial edge of the sample and was named “Cr” (cranial). Subsequently, the diameters of the samples in the dorsoventral (“DV”) and craniocaudal (“CrCa”) directions were measured.

Immediately after sample collection and its measurements, the lateral and deep margins of the specimens were stained with ink and the same measurements were repeated 10 min after incision in all directions with the same method.

After collecting samples from one side of the cadaver, the same procedure was carefully repeated on the opposite side to ensure consistency in data collection.

Following this, the samples were divided into study and control groups, each of which was handled differently according to specific protocols designed to evaluate their distinct characteristics and responses effectively.

The control group samples were immediately fixed in a 10% neutral-buffered formalin solution without any additional support, ensuring straightforward preservation of their anatomical structure.

In contrast, the samples from the study group were fixed onto a 3 mm thick piece of corkboard with the deep fascial plane facing the board utilizing pinpoint needles to secure them in place on a previously drawn 50 mm diameter circle on the corkboard (Figure 2). This method aimed to maintain their original orientation and prevent the distortion of the samples.

After the fixation on the corkboard, the measurements taken for the control group were repeated for the study group in a clockwise direction for the right side’s samples, and in an anticlockwise direction for the left side’s samples.

The study group samples were subsequently fixed in 10% neutral-buffered formalin for a minimum of 48 h to ensure thorough preservation of the tissue architecture. Following this fixation period, the previously described measurements were repeated to evaluate if any changes occurred during this process. To distinguish these new measurements from the initial ones, the letter “F” was appended to the naming convention of each measurement, clearly indicating that they were taken after formalin fixation. This labeling strategy helps in accurately documenting the effects of fixation on the samples and facilitates comparison between pre- and post-fixation data.

After a minimum of 48 h of fixation, the radius of shrinkage was meticulously measured in mm at the dorsal, cranial, ventral, and caudal sites of each specimen. Additionally, the shrinkage diameter was assessed in both the dorsoventral and craniocaudal directions to provide a comprehensive analysis of tissue changes.

All procedures, including specimen drawing, resection, fixation, and subsequent measurements, were conducted by a single resident specializing in small animal surgery. This approach ensured consistency and minimized variability in the handling of samples, thereby enhancing the reliability of the data collected throughout the study.

### 2.3. Statistical Analysis

Statistical analysis was performed with Excel (version 16.0.14527.20162). The Kolmogorov–Smirnov test confirmed the normality of the data distributions. Dimensional differences between the study and control groups were expressed by calculating the means and standard errors of the means. The statistical significance of differences between the two groups was determined by Student’s *t*-test for dependent samples. Data were considered statistically significant when *p* < 0.05.

## 3. Results

A total of 60 samples were carefully collected for the analysis. Each sample had a previously measured radius of 25 mm and diameter of 50 mm documented before excision, ensuring standardized dimensions for subsequent evaluations and comparisons.

No statistically significant difference was noted between the diverse regions of the samples (thorax, flank, femur), and no statistically relevant difference was noted between the measured radii (Cr, Cd, V, D) or the measured diameters (craniocaudal and ventrodorsal) at the different locations. Indeed, all the specimens showed the same trend in every evaluation.

### 3.1. Measurements Directly after Incision

Directly after incision, the study and the control group showed the same trend. The mean radius of all the specimens measured 19.87 mm (with a mean of 19.39 ± 2.3 mm in the study group and a mean of 20.34 ± 2.36 mm in the control group) and the mean diameter measured 40.68 mm (with a mean of 40.07 ± 3.74 mm in the study group and a mean of 41.29 ± 3.68 mm in the control group). This translates to an average reduction of 20.54% in radius and 18.64% in diameter immediately after incision with no significant difference between the two groups.

### 3.2. Measurements 10 min after Incision

Ten minutes after skin incision, the study and the control group showed the same trend. The mean radius of all the specimens measured 18.49 mm (with a mean of 18.38 ± 3.64 mm in the study group and a mean of 18.60 ± 2.38 mm in the control group) and the mean diameter measured 37.22 mm (with a mean of 36.71 ± 4.25 mm in the study group and a mean of 37.74 ± 3.8 mm in the control group). This translates to an average reduction from the original diameter of 26.03% in radius and 25.56% in diameter 10 min after incision with no significant difference between the two groups.

### 3.3. Measurements after Fixation on the Cork Plate

The study specimens were measured once again after they were stretched and fixed onto the cork plate using pinpoint needles and regained the original size. The measurements recorded reflected a mean radius of 24.47 ± 2.23 mm and a mean diameter of 49.55 ± 1.9 mm. This careful process was essential for ensuring that the specimens maintained their anatomical integrity during fixation, enabling accurate comparisons to be made with previous measurements. Such attention to detail is crucial for obtaining reliable data in this type of research.

### 3.4. Measurements after 48 h in Formalin

After 48 h of 10% neutral-buffered formalin’s fixation, the mean radius in the study group measured 25.07 ± 2.19 mm (24.96 mm in thorax samples, 24.94 mm in flank samples, 25.32 mm in femur samples) and the mean diameter measured 49.99 ± 2.35 mm (49.46 mm in thorax samples, 50.24 mm in flank samples, 50.30 mm in femur samples). This translates to an average reduction of −0.28% in radius and 0% in diameter in comparison with the pre-incisional dimensions for the study group (Figure 3a).

After 48 h of 10% neutral-buffered formalin’s fixation, the mean radius in the control group measured 18.57 ± 2.43 mm (18.66 mm in thorax samples, 18.47 mm in flank samples, 18.57 mm in femur samples) and the mean diameter measured 36.84 ± 4.06 mm (36.85 mm in thorax samples, 37.15 mm in flank samples, 36.53 mm in femur samples). This translates to an average reduction of 25.73% in radius and 26.32% in diameter in comparison with the initial pre-incision’s size for the control group. Consequently, a further reduction of 0.14% in radius and 1.79% in diameter was observed after at least 48 h of formalin fixation (Figure 3b).

## 4. Discussion

Multiple factors play a role in the shrinkage process, some of which have not been investigated yet.

The structure and elasticity of every single cell that composes a specific tissue like skin together with its derived tension lines are basic features that can vary at every location on the body. The intrinsic tension of the skin together with the one potentially created by a mass-effect, can change between subjects, breeds, and species. In this regard, feline skin presents denser and coarser collagen bundles, for instance, than canine skin [4], while in dogs, the stratum granulosum directly continues into the lucidum one, differently from cats [5]. Thus, feline skin can more easily shrink. As the results of this study are aligned with previous feline and canine skin studies [3,5], the anatomical difference between these two species does not seem to dramatically influence the overall outcome.

The location of the samples (thorax, flank, femur) was chosen to represent three similar sites to increase the specimens’ number and minimize the tissue’s intrinsic variability. As all the specimens showed the same trend in every analyzed region (thorax, flank, femur) on both sides and for every radius (Cr, Cd, V, D) and diameter (craniocaudal and ventrodorsal) measurement, we can affirm that the location of the sampling did not affect the variability of the results. In fact, no statistically significant difference was observed between the results from the diverse areas of skin sampling.

Initially, both groups showed the same trend in the first moments after incision. In fact, nearly all of the shrinkage took place as soon as the resection began, well prior to formalin fixation. Indeed, in the first minutes after skin incision, a reduction of ca. 20% was already observed in all samples, which increased up to 25–26% ten minutes after specimen removal. Afterwards, the shrinkage remained stable at ca. 26% even after 48 h of 10% neutral-buffered formalin’s fixation. This further slight contraction, which took place during formalin fixation, questionably impacted the overall shrinking. In fact, formalin fixation has for a long time been thought to be the culprit for sample shrinkage. However, several studies in the last few years demonstrated that most of skin shrinkage happens before formalin fixation; hence, formalin does not cause the samples’ reduction [3,5,6,7,8]. Thanks to the findings of our current study, we can also state that formalin does not seem to play an important role in feline skin shrinkage.

The skin reduction observed in our samples by around a quarter from the pre-incisional size has been similarly described in several studies on fresh cadavers in the past years [3,5,6,7,8,9,10]. Risselada et al. for instance described a mean skin specimen shrinkage varying around 21–28% in thirty dogs [3], and a far greater decrease in size in fifteen cats’ skin specimens with a reported value of 58.7% in shrinkage [5]. In this latter study, gelatine spheres were implanted in subcutaneous pockets and subsequently excised to better simulate skin masses or tumors [5]. So far, in most studies published in the literature in veterinary medicine, fresh cadavers have been used. As our findings from thawed animals echo these previous observations and results for both cats and dogs, we assume they reflect an accurate analysis. Skin reduction by ca 25% from the initial size was already described by Risselada et al. as well as by Miller et al. and others among veterinary studies [6,7,8,9,10]. Similar observations have also been found in human studies [7,11].

After 48 h in formalin fixation, all regions from which the specimens were collected showed a significant difference in the measurements between the study and the control groups, confirming our hypothesis. In fact, for all of the control specimens, the size of skin reduction remained stable at around 25–26%, similar to the measurements taken ten minutes after incision and prior to formalin fixation. Instead, an average of 0% in shrinkage was observed in the study group’s samples, where the specimens were stretched and fixed on a cork plate by use of several pinpoint needles. With these results, it appears clear that all specimens of the study group maintained their original pre-incisional dimensions, while the specimens of the control group confirmed the natural tissue shrinkage that occurs after any skin incision. Therewith, the samples’ fixation on a cork plate using pinpoint needles proved to be a valid technique to minimize, if not eliminate, physiological tissue shrinkage.

Despite the numerous studies and research, specimens’ shrinkage remains a highly underexplored phenomenon, and few techniques have been proposed to limit it. Moreover, a valid procedure that reduces specimens’ size and is feasible in the everyday clinical routine has not been described yet. An example of a possible method to stabilize the specimens was given by the study of Risselada et al. published in 2016, in which three suturing techniques between the skin and its fascia were performed during skin sample’s excision in thirty canine cadavers [3]. Unsutured specimens were compared to specimens sutured with a circumferential continuous suture and to further specimens sutured with a four-interrupted quadrant suture. The performance of these diverse suture types on specimen rims proved to decrease both the lateral and the rotational translation of the tissue layers, thus significantly improving alignment of tissue planes without creating distortion. Although good results were obtained to reduce skin layers’ displacement, these methods did not help in avoiding skin shrinkage [3]. They concluded that the placement of rim sutures may be better utilized for tissue samples with thicker deep layers, like complete muscular specimens, and they suggest the use of a simple continuous suture pattern instead of a few interrupted sutures [3]. This study serves as an example to show how complex it can be to effectively reduce or eliminate processes that naturally occur after a tissue incision.

The technique we present herewith allowed for all the samples in the study group to almost completely avoid tissue shrinkage, by restoring the original dimensions with an easy and fast method, and keep these dimensions stable over time. In addition to its seen effectiveness, it should be emphasized that this technique is, first and foremost, easy to understand, fast, and inexpensive. As a matter of fact, this technique does not require any ad hoc material; indeed, any similar alternatives can be considered. Last but not least, no special know-how is needed. Hence, we propose a fast, cost-effective, and easily feasible procedure for skin shrinkage control.

The main limitations of this study stem from the relatively small number of collected samples, which inherently reduces the variability among the examined animals. This limited sample size can impact the generalizability of our findings. Additionally, the intrinsic challenges associated with the use of cadavers must be acknowledged. It is indeed possible that post-mortem changes influenced every measurement taken, especially because of the dehydration and lysis processes that naturally occur after death. These changes can alter tissue characteristics, potentially affecting the accuracy of the results. However, it is noteworthy that these post-mortem alterations were consistently observed across all specimens analyzed. Moreover, these findings were consistent and align well with previous studies reported in the literature, which include both live specimens and fresh feline cadavers [5]. This consistency adds a level of validation to our results, suggesting that, despite the limitations, the study contributes valuable insight into a better and updated understanding of the issue and provides an effective method of control for feline skin shrinkage.

Every cat cadaver showing macroscopic skin diseases or lesions was excluded, but no histopathological examination was carried out to exclude the presence of microscopic skin lesions or changes. This may be a further limitation, as we could not exclude the presence of microscopic changes in the examined skin specimens, which we could not detect during the macroscopical evaluation.

Lastly, all measurements were conducted by a single individual, which typically results in more consistent values. However, this approach may also introduce a greater degree of subjectivity, potentially affecting the overall accuracy and reliability of the results. A balance between consistency and objectivity is essential in this context.

To address these limitations and validate these findings in everyday clinical practice, we plan to conduct additional research through subsequent prospective controlled trials. This approach will help ensure that the results are reliable and applicable in real-world settings, enhancing their significance and utility in patient care.

## 5. Conclusions

The skin samples harvested for our study exhibited notable shrinkage after collection, thus potentially leading to significant discrepancies between the evaluation of surgical and histopathological margins. This can result in inadequate clinical and therapeutical assessments and less accurate decisions regarding the need for or the type of possible treatments. Tissue contraction was evident in all specimens from both groups immediately after incision, starting from approximately 20%, and continued to some extent before formalin fixation, to around 25–26%. This indicates that all tissues shrank by approximately a quarter of their initial dimensions within the first 10 min after cutting. The subsequent different processing of the two groups’ samples revealed a clear disparity after formalin fixation. While the control group exhibited a stable shrinkage rate of approximately 26%, the specimens that were fixed onto the cork plate using pinpoint needles retained their original dimensions measured prior to incision. This observation highlights the effectiveness of this method in preventing physiological tissue shrinkage. Consequently, this simple, fast, and cost-effective technique presents a viable solution for minimizing feline skin shrinkage, making it an ideal approach for researchers and clinicians aiming to preserve the anatomical characteristics and dimensions during specimen analysis to overall improve the reliability and accuracy of patient diagnosis, therapy, and outcome.

## Figures and Tables

**Figure 1 vetsci-11-00507-f001:**
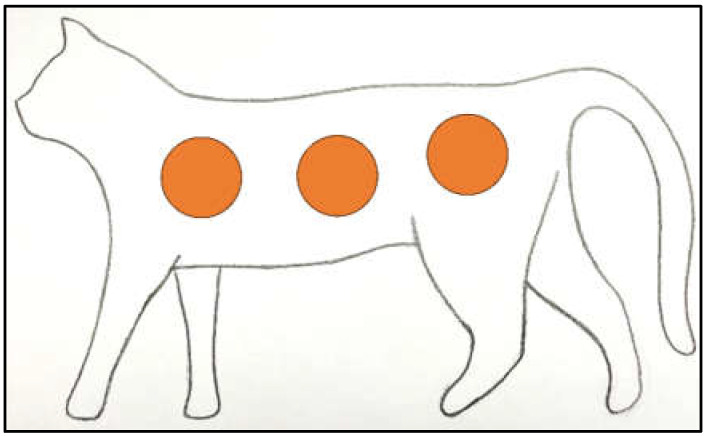
For each cat, six 50 mm diameter specimens from three regions (thorax, flank, femur) were collected.

**Figure 2 vetsci-11-00507-f002:**
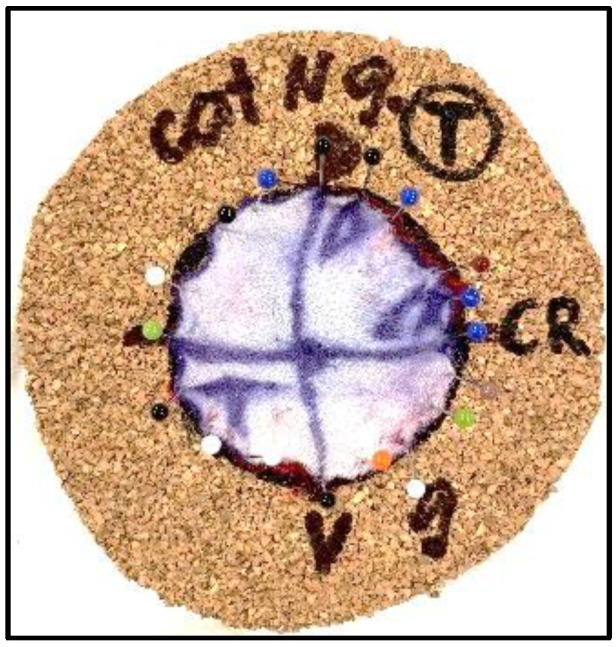
Study group sample stretched and fixed on corkboard with pinpoint needles on a previously drawn 50 mm circle.

**Figure 3 vetsci-11-00507-f003:**
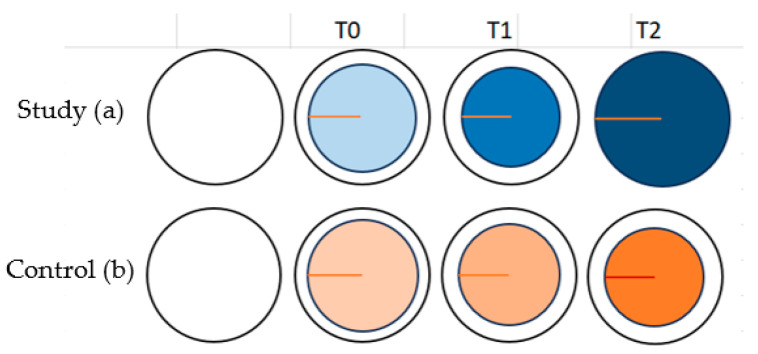
(**a**,**b**) Study and control specimens’ progressive shrinkage—schematic representation before skin incision, directly after incision (T0), 10 min after incision (T1), and 48 h after formalin fixation (T2).

## Data Availability

The original contributions presented in the study are included in the article, and further inquiries can be directed to the corresponding author.

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
