# Peer review of "Anti-Shrinkage Technique for Feline Skin Samples"

_vetsci, 2024, doi:10.3390/vetsci11100507_

Round 1

Reviewer 1 Report

Comments and Suggestions for Authors

The revised article presents a study conducted on feline cadavers with the objective of evaluating skin retraction immediately after excision. Two experimental groups were considered: a control group, where skin segments were immersed in 10% formalin after excision, and an experimental group, where the segments were spread over a cork base and secured with fine-tipped needles to maintain their original dimensions.

This study is both interesting and valuable, providing a practical, effective, and low-cost technique to prevent skin retraction. This allows for a more reliable histopathological examination of the margins and boundaries of the evaluated segments. Such an approach is particularly useful in cases of tumor resection, as it enables a more precise determination of whether the surgical margins are free of tumor cells.

The study addresses a potential solution to the issue of unreliable therapeutic evaluations and imprecise prognoses, which can result from the natural shrinkage of skin samples after collection.

However, several limitations must be considered. Firstly, the small sample size could affect the generalizability of the results. Secondly, the experiment was conducted on cadavers rather than live tissue, which is a significant limitation, as the behavior of live skin tissue differs from that of postmortem tissue, which was also frozen and thawed, leading to cellular rupture. Proper preservation of tissues without cellular destruction requires rapid freezing at temperatures of -80°C or -196°C. However, in this study, cadavers were preserved at -20°C.

Overall, the manuscript is clear, well-structured, and easy to follow. The conclusions are consistent with the evidence and arguments presented. Additionally, the bibliographical references are appropriate and up to date.

Author Response

Comments 1: [However, several limitations must be considered. Firstly, the small sample size could affect the generalizability of the results.]

Response 1: [Thank You for pointing out this aspect. We agree that the small sample size could affect the generalizability of the results. However, these are aligned with several results of previous studies, as stated in lines 189-192 ("manuscript.v5" version). Therefore, we believe our results could apply on a larger scale, but big samples studies are needed to address this limitation.]

Comments 2: [Secondly, the experiment was conducted on cadavers rather than live tissue, which is a significant limitation, as the behavior of live skin tissue differs from that of postmortem tissue, which was also frozen and thawed, leading to cellular rupture. Proper preservation of tissues without cellular destruction requires rapid freezing at temperatures of -80°C or -196°C. However, in this study, cadavers were preserved at -20°C.]

Response 2: [Thank You for this comment and its precision. As also stated in the limitations, we agree with this remark. However, our results are comparable with previous studies on fresh cadavers from Risselada et al. among others (Miller et al., Upchurch et al.), as expressed in lines 189-192 and 212-213 ("manuscript.v5" version). Therefore, we believe these results could reflect an actual situation, but further investigation are needed.]

Reviewer 2 Report

Comments and Suggestions for Authors

Dear Authors,

I really appreciate your tremendous effort you did in this study to submit the manuscript.

By my standpoint, you worked hard but the manuscript to be accepted for publication I consider reasonable just to do a few very minor corrections as I recommended in the attached report.

Author Response

Comments 1 : [Row 83- 84 , please replace „ First, the samples radius was measured in the four directions ”, with „ First, the radius of the samples(skin side outer?) was measured in the four clockwise directions]

Response 1: [Thank You for pointing this out. We agree with the corrections, which make the explanation clearer to the reader. We would then replace it with “First, the radius of the samples was measured from the epidermis side in the four cardinal points in clockwise direction for the right side of the cadavers, and in anticlockwise direction for the left side of the cadavers.]

Comments 2: [Rows 99-100, please precise the samples in the study group were fixed on a 3 mm‐thick piece of cork board with the facial(deep) plane up or skin side up?]

Response 2: [Thank You for this comment. The adaptation expresses the method in a more precise way. We would replace the text with “In contrast, the samples from the study group were fixed onto a 3 mm-thick piece of cork board with the deep fascial plane facing the board utilizing pinpoint needles to secure them in place on a previously drawn 50 mm-diameter circle of corkboard (Figure 2).]

Comments 3: [Rows 109-110, as previously, please precise accordingly how was fixed the sampled on the cork board, how was performed the direction of the radius measurement clockwise or anticlockwise]

Response 3: [Thank You for the comment’s precision. We would replace the text with: “After the fixation on corkboard, the measurements taken for the control group were repeated for the study group in clockwise direction for the right side’s samples, and in anticlockwise direction for the left side’s samples.]

Reviewer 3 Report

Comments and Suggestions for Authors

The limited sample size, and experiments conducted on cadavers may differ from surgical conditions in living subjects. Also, healthy skin tissue is quite different from the skin tissue around subcutaneous tumor. It is recommended that future research should increase the sample size and consider validating these findings in living animals.

In addition,  except fixed on a cork plate, there may be more group for other material, such as hardboard.

Author Response

Comments 1: [The limited sample size, and experiments conducted on cadavers may differ from surgical conditions in living subjects. Also, healthy skin tissue is quite different from the skin tissue around subcutaneous tumor. It is recommended that future research should increase the sample size and consider validating these findings in living animals.]

Response 1: [Thank You for pointing these aspects out. Our data warrant further investigations addressing this study's limitations in controlled larger clinical trials. We are looking forward to proceed with a clinical cases study to further confirm these promising results.]

Comments 2: [In addition,  except fixed on a cork plate, there may be more group for other material, such as hardboard.]

Response 2: [Thank You for Your suggestion. Other materials can for sure be taken into account, as long as they do not compromise the fixation process.]

Round 2

Reviewer 3 Report

Comments and Suggestions for Authors

NA